# Low-Carbon Action in Full Swing: A Study on Satisfaction with Wise Medical Development

**DOI:** 10.3390/ijerph19084858

**Published:** 2022-04-16

**Authors:** Hailin Li, Fengxiao Fan, Yan Sun, Weigang Wang

**Affiliations:** Department of Statistics and Mathematics, Zhejiang Gongshang University, Hangzhou 310018, China; lihailin_1534@163.com (H.L.); ffx562590763@163.com (F.F.); sy23714_zjgsu@163.com (Y.S.)

**Keywords:** low-carbon, wise medical, Zhejiang, co-occurrence analysis, XGBoost

## Abstract

The development of “wise medical” is crucial to global carbon reduction. The medical sector not only has the moral obligation to reduce carbon emissions, but also has the responsibility to provide high-quality services to patients. Existing research mostly focuses on the relationship between low-carbon and wise medical, while ignoring the transformation of wise medical and patients’ opinions in the context of low-carbon transition. The paper crawls the text data of comments on the Zhihu platform (a Chinese platform similar to Quora), explores the focus of patients on wise medical through the co-occurrence analysis of high-frequency words, with a focus directly related to the role of wise medical treatment in carbon reduction, and designed a questionnaire accordingly. Using 837 valid questionnaires collected in Zhejiang Province, an XGBoost model was constructed to discuss the main factors affecting patient satisfaction, and the regional heterogeneity among the coastal area of eastern Zhejiang, the plain area of northern Zhejiang and the mountainous area of southwestern Zhejiang is discussed. The results show that patients’ focus on wise medical lies mainly in the convenience brought by digitalization and the actual medical effect, and the main factors affecting satisfaction with medical treatment are the flow of people in hospitals, optimization of the medical treatment process, the application of digital platforms, the quality of telemedicine services and the appropriate quality of treatment. In terms of regional differences in Zhejiang Province, wise medical is more developed in the plain area of northern Zhejiang, with better simplified medical treatment processes and the construction of a digital platform, while the mountainous areas of southwestern Zhejiang have better quality in telemedicine services despite the geographical environment. Eastern Zhejiang is somewhere in between.

## 1. Introduction

Climate change caused by uncontrolled carbon emissions from human activities has been called “the greatest global health threat in the 21st century” by The Lancet [1]. All countries are currently facing significant and growing adverse impacts of climate change, such as abnormal weather, the spread of disease, and the large-scale movement of climate refugees, which will disproportionately harm the most vulnerable and marginalized populations. It is expected that climate change could push more than 100 million people into extreme poverty by 2030 due to the negative impacts of climate on health [2]. In this context, countries around the world have begun to participate in greenhouse gas reduction activities. The Chinese government’s emphasis on the low-carbon transformation of various industries can be traced back to August 2010, when the National Development and Reform Commission decided to carry out pilot projects for low-carbon industry construction in several provinces and cities [3]. In March 2021, the outline of China’s 14th five-year plan clearly listed “carbon emissions will be stable and moderate after peaking” as China’s long-term goal for 2035 [4].

As one of the fastest growing sectors of the global economy, the carbon emissions of the medical sector account for 5% of total global carbon emissions [5]. If the medical sector were regarded as a country, its carbon emissions would be ranked among the top five in the world. In China, health care recently accounted for 2.7% (68% CI 2.3–3.1) of China’s total GHG emissions [6]. Its carbon emissions account are considerable, but public opinion about carbon emissions often only focuses on traditional manufacturing. In contrast, this data from the medical industry is easily overlooked. Therefore, it is necessary for the healthcare industry to be aware of the impact of climate issues and to be a first mover on the path to carbon neutrality while building resilience and achieving global health goals.

Healthcare industries in each country vary greatly in scale, providing their services while releasing greenhouse gases, purchasing products and technologies from carbon-intensive supply chains. These emissions are focused in infrastructure development, service operations, product supply chains, energy supply, etc. According to statistics, the major contributors of GHG emissions in the Chinese health-care system were public hospitals (148 megatonnes [47%]), non-hospital purchased pharmaceuticals (56 megatonnes [18%]), and construction (46 megatonnes [15%])). It can be seen that the carbon emission problem of China’s medical industry exists mainly in the medical activities of public hospitals [6]. Therefore, in the context of the low-carbon economy, our research focuses mainly on the service system and doctor-patient relationship in public hospitals.

It has become important to transform to a low-carbon economy and form an energy-sustainable industrial development model to mitigate climate change [7]. This is where the environment-friendly model of “wise medical” came into being. The concept of wise medical is also known as WITMED (Wise Information Technology of Med), and can be traced back to the middle of the last century when the telephone became a popular method of communication between doctors and patients [8]. It can be divided into two stages; first, at the end of the last century, with the development of information technology and semiconductor technology, great progress has been made in the development of telemedicine [9]. Second, in recent years, the popularization of mobile terminals and the Internet has promoted telemedicine’s entry into a new stage of medical information interconnection [10]. At this stage, a large number of emerging wise medical technologies are gradually maturing, such as big data-driven decision-making models, medical artificial intelligence [11], artificial intelligence wearable devices based on IoT sensor networks [12,13,14], etc. In China, the concept of wise medical is closely linked to the “new medical reform” policy of 2009. So far, Chinese academic circles have placed different emphasis on the concept of “wise medical”, but we conclude that “wise medical” is mainly based on the Internet of Things, cloud computing, data mining, etc., as its technical basis, taking the patient as the center, promoting comprehensive informatization of medical and health services, realizing the interconnection between patients and medical personnel, medical equipment, and medical institutions, so as to improve the efficiency of medical services [15], which means simplifying bloated healthcare facilities, and reducing waste of resources and unnecessary carbon-emitting activities. In the context of the re-emergence of COVID-19, various regions in China are in a relatively closed state to minimize the flow of people. The advantages of comprehensive informatization and the real-time data updates made possible by wise medical are particularly important at this time, but from the perspective of their responses, the public’s satisfaction with its effect is still mixed.

Wise medical directly shortens travel distance, reduces diagnosis time and treatment time, effectively limits crowds gathering during the epidemic, and promotes the sharing of high-quality medical resources. It reduces the workload of hospitals to a certain extent, and makes an important contribution to carbon reduction. However, when the medical industry transforms to wise medical, although it can effectively reduce carbon emissions, the rapid process of efficiency improvement of medical resources may reduce patient satisfaction to a certain extent [16]. For example, changes in the energy structure within the hospital affect the experience of medical treatment, and low quality telemedicine services lead to inaccurate diagnosis results.

According to our survey, research on the relationship between wise medical and carbon emissions in China is very scarce, and more research is required on the efficiency improvements and service optimization brought by wise medical. However, Rui Wu studied life cycle GHG emissions of the Chinese health-care system [6], and obtained the macroscopic status of carbon emissions of various areas in Chinese medical institutions. This was an important literature basis for our decision to conduct research into the impact on service experience of wise medical systems in public hospitals. There are also relatively few related studies abroad, mainly concerning measurement of the carbon footprint of medical activities or individual medical institutions. Typical examples include the carbon footprint study of Australian medical services by Malik et al. [17], and measurement of the carbon footprint of the Health Service (NHS) [18]. Forner measured carbon footprint savings associated with surgery outreach clinics by tracking and surveying patients’ travel activities for medical treatment [19], and in the context of the epidemic, there is comparative study of the carbon footprint of a Geriatric Medicine clinic before and after COVID-19 [20]. In addition, Purohit came to the conclusion through a systematic review that telemedicine does reduce the carbon footprint of healthcare, primarily by reduction in transport-associated emissions [21]. According to these findings we determined the research premise that wise medical has a positive effect on carbon reduction. Therefore, based on the focused areas of Chinese academia and the existing literature, and on the premise of the research conclusions of foreign scholars on medical carbon emissions, this paper studies the medical experience brought by the development of wise medical for patients in the context of a low-carbon economy, as well as the follow-up development of wise medical construction.

Taking Zhejiang Province as a case, this study used 837 questionnaires obtained from the survey to explore patients’ satisfaction with the use of wise medical against a low-carbon background. First, based on online comments on the Zhihu platform, we constructed a semantic network of co-occurring words to screen relevant indicators. These indicators are closely related to patients’ experience in the low-carbon construction of hospitals, and we further used XGBoost to study the importance of these indicators affecting residents’ medical experiences. 

Finally, the province is divided into the coastal area in eastern Zhejiang, the plains area in northern Zhejiang, and the mountainous area in western Zhejiang. The respective level of construction of wise medical in the three geographical divisions and the regional heterogeneity are studied, and the future development direction is discussed.

## 2. Materials and Methods

### 2.1. The Study Area

Zhejiang Province is located in a developed area along the eastern coast of China, with outstanding medical resources and scientific innovation capabilities. It is one of the pilot provinces for comprehensive medical reform and one of the demonstration provinces for “Internet Plus Medical Health” in China. However, in the context of global warming, Zhejiang Province, a major source of carbon emissions in China, has seen an accelerated trend of rising temperatures and an increase in the frequency of meteorological disasters such as typhoons, rainstorms and floods, extreme high temperatures and droughts. In 2019, the sea level in Zhejiang Province was 93 millimeters higher than the average year-on-year, which was the second highest level since 1980. Therefore, the province is also a leading demonstration province to actively implement the “14th Five-Year Plan” to address climate change, and is in a critical period of low-carbon social transformation. From the perspective of medical and health care, from 2013 to 2020 the scale of health institutions in Zhejiang Province fluctuated and increased. In 2020, Zhejiang Province had a total of 34,400 health institutions (including village clinics) and 548,000 health technicians, ensuring sufficient observation samples for investigations. At the same time, the huge medical scale has boosted the informatization construction of intelligent medical care: in 2020, the number of appointment requests on the Zhejiang diagnosis and treatment service platform reached 10.49 million, and the number of successful appointments exceeded 7.96 million, year-on-year increases of 3.8% and 8.2% respectively, and a total of 1901 medical and health institutions were connected. In terms of policy direction, in May 2020, the Zhejiang Provincial Health and Health Commission issued the “Internet Plus Medical Health” Demonstration Province Construction Action Plan (2019–2022); its goal requires that by 2022, 80% of the basic data of medical and health institutions can be shared and applied, and the comprehensive digital transformation of health services and governance will basically be realized.

### 2.2. Basic Data and Processing

The questionnaire data used in this paper were collected from an online survey in December 2021. The survey location was Zhejiang Province, divided into the coastal areas of eastern Zhejiang (Shaoxing; Ningbo; Zhoushan), the plains areas of northern Zhejiang (Hangzhou; Jiaxing; Huzhou), and the mountainous areas of southwestern Zhejiang (Wenzhou; Taizhou; Lishui; Quzhou; Jinhua) based on geographical divisions [22]. A total of 902 questionnaires were distributed to the three divisions through multi-stage stratified sampling, and 837 valid questionnaires were obtained after eliminating those whose answers were too rough and casual, those where important items were missing, and those from respondents who had not been hospitalized.

According to statistics, the ages of the respondents were mainly distributed in the ranges of 22–30 years old, 31–40 years old and 41–60 years old, with respective proportions of 27.1%, 36.0% and 29.1%. In the valid samples, the proportions of males and females were 52.09% and 47.91%, respectively, which was relatively balanced. At the same time, these respondents all had certain hospitalization experience and good understanding of local hospitals, so meeting the research needs.

### 2.3. Methods

#### 2.3.1. Selection of Indicators Affecting Wise Medical

Wise medical mainly takes medical information as its main focus, and uses information interaction, network communication and other technologies to achieve close interaction between patients, medical equipment, medical institutions, and medical personnel, thereby establishing an efficient and scientific medical service system [23], increasing the convenience of seeking medical treatment [24]. It tries to improve the medical experience in many ways and to reduce resource waste and energy loss, but the construction and application of wise medical is not yet mature, and may also bring privacy and security issues [25] and difficulties learning equipment operation [24]. Therefore, patients have varying experiences with wise medical, and specifically the issue of patient satisfaction needs to be further explored.

Wise medical assumes the responsibility of the medical industry itself, that is, providing benefits for patients seeking medical treatment, and also takes into account the important responsibility of energy conservation and emissions reduction. Therefore, two conditions had to be met in the process of indicator selection: (a) the indicator itself is directly related to low carbon, and refers to the transformation from traditional medical care to wise medical; (b) the indicator is closely related to the interests of patients, and refers to patient concerns during medical treatment processes. In view of the regional differences in the development of wise medical, this stud crawled patients’ online reviews from the Zhihu platform to construct a corpus, and used co-occurrence analysis to mine patients’ concerns about wise medical.

The co-word network is an undirected weighted network whose main goal is to analyze the relationship between words in the text [26] and establish the network topology according to the contribution of keywords. In the co-word network, association strengths between words are not completely equal. The more frequently two words appear together in different reviews, the stronger their association [27]. Existing research points out that co-word networks have advantages over traditional text analysis in terms of theoretical construction [26]. Therefore, it is applicable to use the co-word network to analyze the concerns of patients about the construction of wise medical care (Figure 1).

#### 2.3.2. Measurement of Indicators Affecting Wise Medical

XGBoost is an ensemble algorithm based on gradient boosting trees. It constructs several weak evaluators by introducing a regularization loss function into the model, and integrates these weak classifiers with lower accuracy into a strong classifier with higher accuracy, which not only reduces the risk of model overfitting, but also makes its classification performance better than a single model [28]. Since XGBoost uses technologies such as pre-sorting, weighted quantiles, sparse matrix identification, and cache identification [29], it has the advantages of parallel operation, controllable algorithm complexity, and strong generalization ability. Its objective function is
(1)$$Obj=\overset{n}{\underset{i=1}{\sum }}l\left(y_{i},\hat{y_{i}}\right)+\overset{k}{\underset{k=1}{\sum }}\Omega \left(f_{k}\right)$$
where *l* represents the loss function, *y_i_* represents the true value of the *i*-th surveyed *x_i_*, $\hat{y_{i}}$ represents the predicted value of the *i*-th surveyed *x_i_*, and *f_k_* represents the prediction function of the *k*-th tree.

While classifying predictions, the XGBoost model can also calculate the feature importance, which is used to measure the influence of an indicator on the label variable. There are three main methods for calculating feature importance: gain, weight, and cover. The importance was calculated in this study using the method of weight, which interprets the importance score as the number of times each feature appears in the boosted tree, that is, the number of times the feature is used as a split node [30]. Due to the difference in the degree of use and sensibility of patients in medical services, the universality and influence of different aspects of medical service quality were divided into primary and secondary. The score was used to measure the impact of each wise medical impact indicator on the final overall satisfaction; the higher the score, the more important the impact of the indicator on the overall satisfaction with the hospital. Therefore, it is the area that highlights the main patient experience differences in many aspects of service quality.

#### 2.3.3. Measurement of Level of Wise Medical

After finding the main factors that affect the patient’s medical experience in the construction of wise medical, it was necessary to evaluate the construction level of these aspects to obtain the main needs of patients. Generally speaking, the construction level of wise medical should be objectively evaluated from the data of hospital construction, but it may be difficult to obtain data, and the number of hospitals was large, so this was difficult to obtain one by one. Thus, we chose to reflect the construction level of wise medical through the questionnaire of the respondents, that is, through the score of satisfaction:(2)$$C_{nj}=\overset{m_{n}}{\underset{z=1}{\sum }}x_{nzj}/m_{n}$$
where *C_nj_* represents the construction level of the *j*-th indicator in the *n*-th region, *m_n_* represents the number of respondents in the *n*-th region, and *x_nzj_* represents the satisfaction of the *z*-th respondent in the *n*-th region with the *j*-th indicator.

## 3. Rusults

### 3.1. Selection of Indicators

This paper used wise medical as the keyword to search, crawled the relevant comments of China Zhihu (https://www.zhihu.com/ accessed on 9 January 2022) from 2015 to 2021, and preprocessed the acquired text data. The processing process was as follows: First, regular expressions were used to filter special characters in the text, combining the “jieba” word segmentation package in Python to segment the Chinese text. Then, we comprehensively considered “Harbin Institute of Technology Stopword List”, “Chinese Stopword List”, “Sichuan University Machine Intelligence Laboratory Stopword List” to remove stop words from the segmented text. It is worth mentioning that in the process of word segmentation, a customized dictionary based on wise medical professional vocabulary was added to improve the word segmentation effect. Meanwhile, Chinese words less than one characters (useless to our research) and traditional hospital-themed words such as “patient” and “nurse” were removed to reduce the impact on the final analysis. The top 30 words that appear most frequently are shown in Table 1.

High-frequency words reflect to a certain extent the focus of Zhihu users on wise medical. Overall, the focus is on the convenience brought by wise medical services. However, some high-frequency words which do not fully reflect the development characteristics of wise medical were found to exist independently. Therefore, further co-occurrence analysis was needed to seek the relationship between different keywords. Figure 2 shows the co-occurring word semantic network constructed in this paper.

It can be seen from Figure 2 that the three keywords of telemedicine, service and intelligence define the discussion framework of the existing wise medical system. At the same time, these three high-frequency words also reflect the overall characteristics of wise medical in Zhejiang Province. From the perspective of telemedicine, wise medical transfers part of the medical treatment process online through digital means, such as online appointments, autonomous operation of machines and equipment, electronic reports, electronic medical records, etc. This model can reduce the flow of people in hospitals and shorten patients’ travel time, thereby reducing carbon emissions; from the perspective of service, the wise medical model includes scientific planning of department divisions in the hospital structure, optimizing the offline medical treatment process. With strengthened management of wards and medical staff, the comfort of the patient’s medical experience can be improved, and their recovery time can be shortened. At the same time, the energy required for hospital operation, such as hot water supply, canteen catering insulation, etc. has also been optimized in the energy structure and can meet low-carbon standards. From the perspective of the keyword wisdom, this invokes a combination of many wise medical features, and is closely connected with electronic means and medical service systems. Digital medical means can provide many online platforms, give patients more information and convenience, and enable patients to access more medical resources at home. At the same time, wisdom also means scientific management of the above resources and natural resources through participatory management [31].

Based on the above analysis, this paper gives the relevant indicators that affect the satisfaction of wise medical use. The specific indicators and descriptions are shown in Table 2. Our scale is based on this design.

### 3.2. Research on Influencing Factors Based on XGBoost

#### 3.2.1. Reconstruction and Division of Samples

According to the original survey data, the proportion of respondents who were satisfied or dissatisfied with the overall tendency of the hospital was 697:140, indicating a certain sample imbalance. To solve problems of unbalance, threshold shifting, oversampling, undersampling [32], or building a cost-sensitive function may be applied [33]. After the actual test, resampling was able to achieve better results, and the overall tendency of the sample to be reconstructed became 697:697. After the sample reconstruction was completed, the samples were divided according to a ratio of 7:3, that is, the training set accounted for 70% and the test set accounted for 30%. The XGBoost model was constructed with the patient’s overall satisfaction tendency *Y* as the classification feature and *X*_1_–*X*_14_ as the characteristic variables.

#### 3.2.2. Parameter Tuning

The choice of parameters has an important impact on the results of any machine learning model [34], and inappropriate choices may lead to problems such as overfitting and underfitting. Therefore, in order to improve the model’s accuracy and generalization ability, the parameter tuning process is indispensable. In this paper, algorithm parameters were tuned based on K-fold cross-validation and grid search. The optimal parameter selection is shown in Table 3.

The XGBoost model required the adjustment of many parameters and indicators, such as n_estimators, subsample, and learning rate, representing the number of iterations, the random sampling ratio, and the learning rate, respectively [35]. The precision, recall, and f1-score of the final model are all 0.85 (see Table 4 for details), so the model has a good fitting effect.

#### 3.2.3. Feature Importance Ranking

After completing sample reconstruction, sample division, and parameter tuning, the importance ranking of each feature was obtained. The results are shown in Figure 3.

As can be seen in Figure 3, in the context of wise medical, the optimization of the flow of people and the process of seeing a doctor within the hospital has the greatest influence on the patient’s medical experience. In fact, an important way to reduce carbon in wise medical is to optimize the internal structure of the hospital to reduce the stay time of patients in the hospital, and thus achieve the goal of carbon reduction, namely, structural carbon reduction [36]. Healthcare is economically, ethically and financially obligated to achieve better health services, and also to lead processes of carbon reduction [37], more so now than ever. With a sharp increase in population, medical resources become more scarce, but improvements in efficiency and the optimization of the structure can make up for the shortage in volume, and release more abundant medical capabilities. The digitalization and telemedicine service levels are less important than the preceding two, but are still significantly larger than other characteristic variables. 

An important feature of wise medical is the involvement of the Internet and big data. A digital medical system can greatly improve the efficiency of the entire medical process and achieve the purpose of reducing carbon emissions, especially when electronic equipment is popularized. Therefore, hospital digital platform construction is extremely important for patients. At the same time, telemedicine services are a derivative of the construction of digital platforms, which can directly reduce carbon emissions generated during patient travel. For patients in remote mountainous areas, telemedicine is even more important [38], and overall it is an indispensable part of improving the medical security system in China. 

In addition, excessive medical treatment has always been a chronic problem in the medical service system. Due to the information asymmetry between doctors and patients, doctors may perform unnecessary medical actions on patients, especially in mild cases, increasing the economic burden on patients [39]. Therefore, the importance of the moderation of medical behavior is high, which reflects the patient’s expectation of reducing excessive medical behavior in the context of wise medical care. The other characteristics also have a certain impact on patients’ satisfaction with medical treatment, but their importance is significantly lower than the first five. This set of characteristics include the construction of infrastructure, professional equipment, layout planning and service capabilities. Wise medical not only reduces carbon emissions and promotes the convenience of medical treatment, but may also increase the patient’s medical expenses, which is precisely the negative impact that wise medical may bring. The implementation of a wise medical system requires the use of new technologies, which will increase the cost of doctors’ training and may reduce the accuracy of online medical treatment. Therefore, patients not only pay attention to the benefits brought by wise medical, but also pay attention to the possible negative effects. These are also key considerations in the wise medical construction process.

### 3.3. Regional Heterogeneity of Wise Medical Construction

Since the importance of the first five indicators is obviously stronger than that of other indicators, only the first five indicators are discussed at the level of wise medical, and the calculation results are shown in Figure 4.

In terms of the optimization of the medical treatment process, overall satisfaction in the northern Zhejiang plains is higher than in the eastern Zhejiang coastal areas, while this indicator in southwestern Zhejiang is less than 3, much lower than 3.89 in the northern Zhejiang plain and 3.5 in the coastal area of eastern Zhejiang, showing a relatively large gap. The optimization of the diagnosis and treatment process can reduce registration and other processes, so it can simplify department structures and identify redundant human resources, to achieve the purpose of reducing carbon emissions. The quality of the simplification of treatment processes is related to the implementation of medical reform policies and the service level of the hospital itself. Southwestern Zhejiang is geographically far away from the political center of the provincial capital, and the efficiency there of medical reform and the simplification of processes is observably lower than in northern and eastern Zhejiang. From the perspective of the geographical distribution of medical resources, there are significant regional differences in the distribution of medical and health resources in Zhejiang. The overall trend is a gradual decrease from coastal cities to inland cities. High-quality medical and health resources are highly concentrated in the northern Zhejiang plain [40], especially in Hangzhou. There is a relative gap in the management level of hospitals in the southwestern region, so there is also a gap in the efficiency and thoroughness of process simplification linked to it.

As far as the construction of digital platforms is concerned, the geographical distribution of satisfaction with the optimization of the medical treatment process is follows the same pattern. Northern Zhejiang outperforms eastern Zhejiang, which in turn performs better than southwestern Zhejiang. The reason is that the construction and popularization of digital information platforms can positively promote medical treatment process optimization. For example, through the Zhejiang Provincial Hospital Appointment and Diagnosis Platform, patients in the countryside or the city can make an appointment directly with the provincial hospital for examination and hospitalization, eliminating the complicated intermediate registration process previously used. This thereby reduces carbon emissions in the user’s medical diagnosis and treatment process, so there exists geographical correlation to certain extent. The reasons for this geographical distribution also include the degree of implementation of medical reform policies, and the uneven distribution of high-quality medical resources. The construction of and access to digital platforms are closely related to the level of scientific and technological development and economic development in each region. Naturally, the economy of Zhejiang Province shows synchronous differences in regional distribution.

In terms of the flow of people in the hospital, Eastern Zhejiang performs better than both northern and southwestern Zhejiang. The intensity of the flow of people in hospital can be regarded as the result of the comprehensive influence of the construction of the digital platform, the optimization of the medical treatment process, and the quantity of telemedicine services. The higher the degree of informatization, the better the simplification of the process; the higher the satisfaction with telemedicine services, the lower density of people in the hospital, which will reduce the carbon emissions of patients in the hospital. Therefore, in southwestern Zhejiang, where the satisfaction with digital platform construction and process simplification is far behind, hospitals show a high degree of crowd density. At the same time, the density of people in the hospital is directly related to the population density and population size of the region. Although information construction and process optimization in eastern Zhejiang is slightly inferior, there are larger, more densely populated cities in northern Zhejiang, and the total number of patients and their demand for medical treatment is higher than that in the coastal areas of eastern Zhejiang. Therefore, the satisfaction level for the internal flow of people in hospitals in eastern Zhejiang is better than that in northern Zhejiang.

When it comes to the quality of telemedicine services, the most significant feature is that satisfaction with all other indicators is relatively backward in southwestern Zhejiang, but there also the satisfaction with this indicator is the highest. We believe that this is related to the strong demand for telemedicine services in southwestern Zhejiang Province. Compared with the plains of northern Zhejiang and the coastal areas of eastern Zhejiang, the geographical environment of southwestern Zhejiang is more mountainous and hilly, and there is a gap in the geographical distribution of high-quality medical resources. The telemedicine service platform allows doctors in grassroots hospitals to provide medical services directly from their workstations. Ordinary people can make appointments for specialist outpatient clinics in provincial hospitals by referral, so patients in the community can enjoy the medical resources of provincial tertiary hospitals, and promote carbon reduction by lowering traffic travels. Therefore, people in this area have a stronger willingness to use and more intense demand for telemedicine services. In eastern and northern Zhejiang, where digital construction and process optimization efficiency are high, the difficulty accessing local high-quality medical resources is very low, so patients there are more concerned about the cost of telemedicine services and the accuracy of diagnosis.

The satisfaction of the appropriate quality of treatment in the three places is basically the same, all at around 3.1. We believe that this indicator reflects the relatively high overall awareness of medical services in Zhejiang Province, suggesting that modern wise hospitals not only develop in a low-carbon direction, but also pay attention to the interests of patients and the code of conduct for medical staff.

## 4. Discussion

### 4.1. A Textual Co-Occurrence Network Analysis of Patient Concerns

In this study, we used a text co-occurrence network to analyze netizens’ comments on hospitals’ informationization, to capture the public’s direct feelings and their attitudes to services in the process of medical low-carbon transformation. We extracted the three core nodes of telemedicine, service and wisdom. In accordance with most of the existing research, we also believe that the public has a unique preference for the convenience and benefits of wise medical. In fact, with the COVID-19 pandemic, telemedicine is gaining popularity [41,42] to assist with post-treatment monitoring of recovering COVID-19 patients, to reduce the number of frontline workers being infected during the pandemic [43]. However, there remain obstacles to this method of medical treatment, including technical obstacles, regulatory obstacles, financial and organizational obstacles, etc. [44]. Therefore, strengthening the maturity and service level of telemedicine services is also a very important part of the later construction of wise medical. This conclusion is consistent with the outcome from the XGBoost model training, indicating that the telemedicine service level feature has a high importance. 

However, according to our observations, most other scholars consider only the needs of doctors or patients for convenience when selecting indicators [45,46], showing lack of consideration for carbon emissions. This approach may ignore the origin of the values of wise medical care, and focus only on human experience without paying too much attention to changes in the environment, which is inappropriate. The development of wise medical care is inseparable from the need for carbon reduction. In the indicators we constructed, the key word extended from the node of wisdom illustrates the embodiment and application of wise medical care in carbon reduction. We considered the aspects by which wise medical can promote carbon reduction to measure the actual experience of patients in the process of low-carbon transformation of hospitals, which can be regarded as innovative application of text mining to improve the objectivity and scientificity of the index system construction process.

### 4.2. Ranking of Satisfaction Factors Based on XGBoost

Patient satisfaction with medical treatment is an important indicator to measure levels of medical service [47]. From the perspective of patient satisfaction with medical treatment, many scholars have studied the factors that affect patient satisfaction from the perspective of traditional medical care, and pointed out that patients’ medical satisfaction is related to medical care provision [48], number of hospital beds [48], privacy protection [49], etc. These factors are essentially the service level that patients receive during the medical treatment process, which is basically consistent with our research results, that patients have high requirements for quality service level and efficiency of seeing a doctor.

In our research on influencing factors, although lighting systems, equipment operation convenience, catering, etc. are also in patients’ focus, their feature importance is relatively low, showing that patients care most about medical care level, which is in line with their fundamental interests. However, hospitals also have moral obligations; the follow-up research should not only focus on the decarbonization path of wise medical [37,50], the construction system of wise medical [37] and the development of wise medical [51,52,53] to promote environmental harmony, but also how to improve the comfort of patients during medical treatment, although these matters are felt to be less important than the quality of medical care.

### 4.3. Analysis of Regional Heterogeneity

For the evaluation of the development level of wise medical, the existing research has been carried out mainly from the two perspectives of hospital internal hardware equipment [54] and statistical measurement [55,56]. These studies may be comprehensive in dimension, but from the perspective of patients, these research results are numerically absolute. We should take into account the actual needs of local patients to evaluate the relative construction of wise medical [57]. At the same time, patients’ requirements for the quantity and quality of medical service resources vary between regions with different economic levels and population conditions, that is, regional heterogeneity [58]. The degree of medical intelligence and the status of informatization construction vary between regions, and there are differences in patients’ attitudes and acceptance of low-carbon transformation of hospitals. These differences are related to, for example, requirements for the accuracy and safety of telemedicine services [25,46], the different learning costs of wise medical, and the depth of use of digital medical information platforms [46]. These heterogeneities mean that the construction of wise medical has different focuses in different times and places. In this regard, the relationship between supply and demand is more complicated. We divided Zhejiang Province into three areas: the plains area of northern Zhejiang, the coastal area of eastern Zhejiang, and the mountainous area of southwestern Zhejiang, and directly used the subjective perceptions of patients to evaluate the construction of local wise medical. The method was simple, but it may be efficient. Compared with existing studies [59,60], this study can better identify the construction of wise medical in different regions of Zhejiang Province, and effectively guide the construction priorities of each subregion in the later stages.

### 4.4. Limitations and Prospects

Firstly, evaluation of the construction level of wise medical is represented by the average values of patients’ satisfaction. While patient evaluations may be more accurate than some methods, and prevent hospitals from exaggerating their features for reputational reasons, the practice is still somewhat subjective. If resources are sufficient, statistical measurement should be carried out within the hospital construction department to quantitatively evaluate the construction situation of wise medical in all its aspects. 

Secondly, based on the traditional concept of carbon emissions, we presumed that the informatization of healthcare will have a positive emission reduction effect. In fact, the relationships between sources of carbon emissions involved in the wise medical system is more complicated, and include carbon emissions caused by electronic equipment itself. However, the data required to accurately measure the emission reduction effect have been ignored by hospitals and other healthcare institutions, so it is difficult to collect and conduct in-depth and specific investigations. 

Thirdly, we studied Zhejiang Province, a relatively developed area of China in terms of economy and medical resources. Due to China’s regional characteristics and complex social structure, the wise medical system has not yet been popularized and developed in poorer areas. The potential of generalization is limited, but the relationship between wise medical and patient satisfaction obtained by our analysis of regional heterogeneity is of general significance. Fourthly, although the satisfaction of patients is mostly based on the static experience of local medical provision, and they lack relative awareness of regional heterogeneity, this corresponds with the existence of relative isolation between regions under the current background of the re-emergence of COVID-19 in China. At the same time, we believe that in this context, the regional heterogeneity of patients’ experience of wise medical in this study can be used as a basis to further explore how wise medical can break the occlusion of medical information, and the imbalance between supply and demand for medical resources in the epidemic environment.

## 5. Conclusions

We focused attention on the relationship between the construction of wise medical and the patient experience in the low-carbon transformation of hospitals. Taking Zhejiang Province as an example, this paper conducted a survey and research into patient satisfaction with wise medical from the outside to the inside, and in the process of analysis discussed the reasons behind the results and the deviation of wise medical development caused by geographical factors.

The results show that residents’ focus on wise medical mainly lies in the convenience of medical treatment and the actual treatment effect brought by the new wise medical model. Research on the factors influencing these main aspects was carried out, and it was found that the top five main factors affecting overall satisfaction of patients in medical treatment are the flow of people in the hospital, the simplification of the medical treatment process, the application of the hospital’s digital platform, the quality of telemedicine services, and the appropriate quality of treatment. In addition, for Zhejiang Province, there have been significant differences in the construction of wise medical in the coastal area of eastern Zhejiang, the plains area of northern Zhejiang, and the mountainous areas of southwestern Zhejiang. The plains area of northern Zhejiang has more high-quality medical resources, and the level of wise medical construction is relatively high. The construction of digital medical care is significantly higher than that in other regions of Zhejiang Province, roughly in line with the level of economic development. Although the overall development of the mountainous areas in southwestern Zhejiang is weaker than that of other areas in Zhejiang Province, the quality of telemedicine is relatively high, which is driven by the fact that it is difficult to see a doctor in the mountainous areas.

## Figures and Tables

**Figure 1 ijerph-19-04858-f001:**
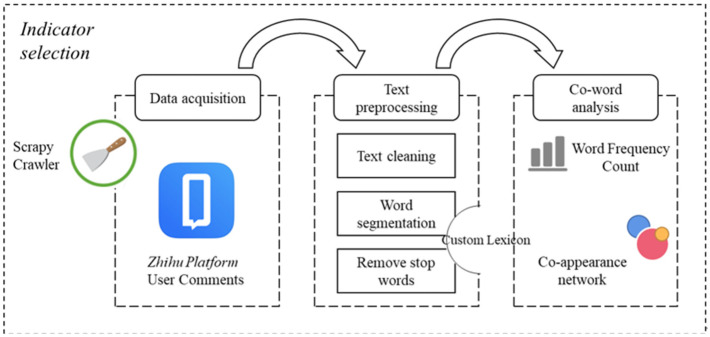
The process of indicator selection. This figure shows the whole process of indicator selection in this paper. After crawling the text, this study processes the patient comments through three text preprocessing methods, and then analyzes it through the co-word semantic network.

**Figure 2 ijerph-19-04858-f002:**
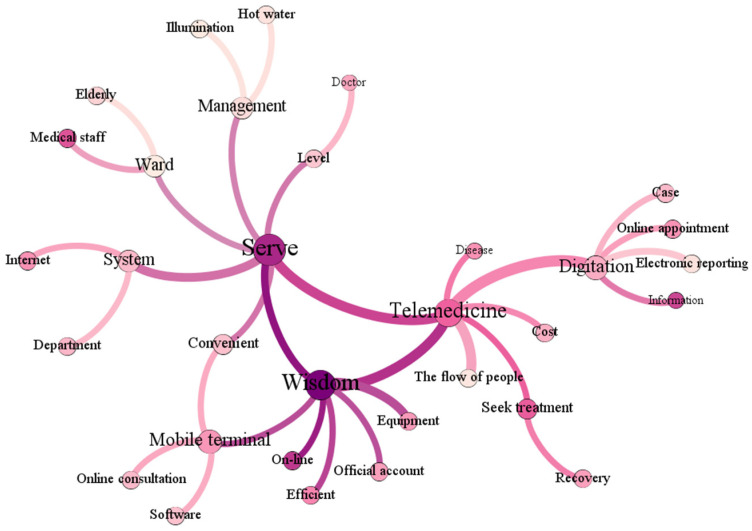
Co-occurrence analysis of high-frequency words in wise medical literature.

**Figure 3 ijerph-19-04858-f003:**
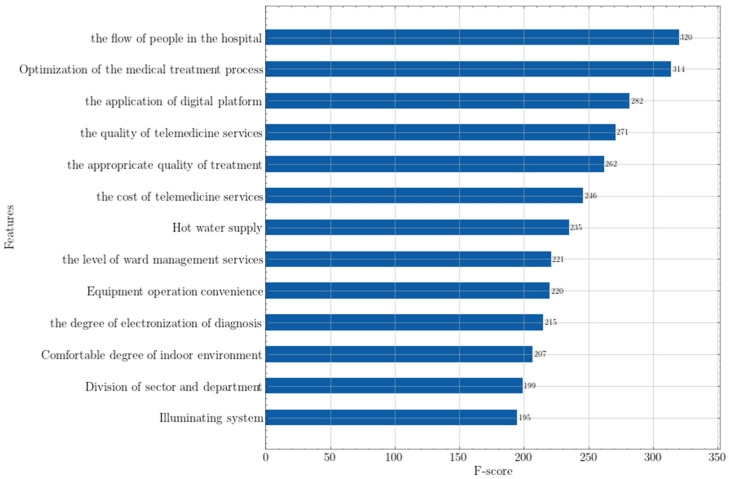
Ranking of feature importance in XGBoost. The score of feature importance only has relative significance, and does not numerically reflect the multiple relationships of importance between indicators.

**Figure 4 ijerph-19-04858-f004:**
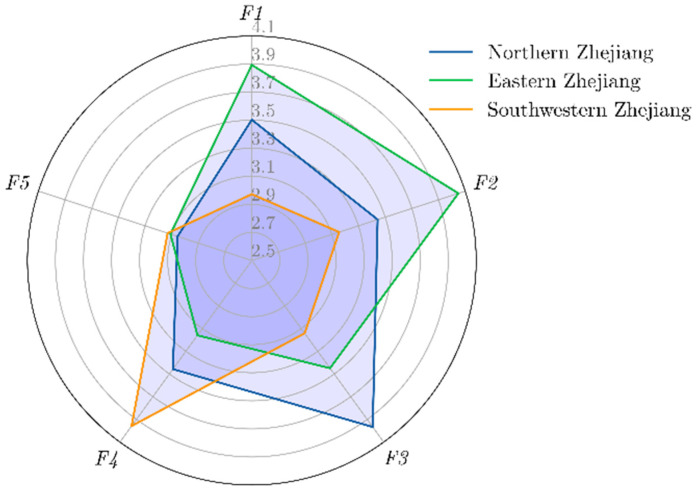
The construction level of wise medical in various regions of Zhejiang Province. *F*_1_, *F*_2_, *F*_3_, *F*_4_, and *F*_5_ represent the optimization of the medical treatment process, the construction of digital platforms, the flow of people in hospital, the quality of telemedicine services, and the appropriate quality of treatment, respectively.

**Table 1 ijerph-19-04858-t001:** High-frequency words in comments from the Zhihu platform.

Keywords	The Number of Occurrences	Appearance Ratio	Keywords	The Number of Occurrences	Appearance Ratio
Wisdom	545	0.0401	Digitation	215	0.0158
Serve	464	0.0341	Cost	211	0.0155
On-line	427	0.0314	Department	199	0.0146
Information	400	0.0294	Online consultation	193	0.0142
Medical staff	368	0.0271	Case	193	0.0142
Seek Treatment	349	0.0257	System	192	0.0141
Telemedicine	343	0.0252	Convenient	182	0.0134
Efficient	280	0.0206	Software	181	0.0133
Internet	260	0.0191	Level	170	0.0125
Disease	258	0.019	Elderly	152	0.0112
Online appointment	258	0.019	Management	139	0.0102
Mobile terminal	250	0.0184	Electronic reporting	128	0.0094
Equipment	245	0.018	Hot water	125	0.0092
Official Accounts	235	0.0173	Flow of people	118	0.0087
Recovery	235	0.0173	Illumination	114	0.0084

**Table 2 ijerph-19-04858-t002:** Indicators influencing wise medical, and what they describe. *Y* is a binary variable of 0–1, representing dissatisfaction and satisfaction, respectively; *X*_1_–*X*_14_ are ordinal variables, with values 1–5, representing very dissatisfied, relatively dissatisfied, average, relatively satisfied, and very satisfied, respectively.

Indicators	Symbol	Description
Overall satisfaction tendency	*Y*	Describes patients’ overall satisfaction tendency with the first-line hospital when seeking medical treatmentin a low-carbon context.
Equipment operation convenience	*X* _1_	The machinery and equipment in the hospital is an important feature of wise medical. This indicator describes the degree of patients’ satisfaction with the convenience of the operation, and the hospital’s machinery and equipment.
Optimization of the medical treatment process	*X* _2_	A perfect medical procedure should be able to increase the efficiency of the patient’s medical treatment. This indicator describes patients’ satisfaction with the hospital’s medical procedure.
The degree of electronization of diagnosis	*X* _3_	Including payment, physical examination report, medical record book, etc., reflecting the degree of “paperlessness” in the process of medical treatment
Illuminating system	*X* _4_	Energy supply structure is an important way for hospitals to reduce carbon, reflecting the satisfaction of patients with hospital lighting systems under the new energy supply.
Division of sector and department	*X* _5_	The scientific department location distribution can speed up the process of seeing a doctor and reduce the patient’s length of stay in hospital. This indicator reflects patients’ satisfaction with the rationality of department partition.
Hot water supply	*X* _6_	Reflects the satisfaction of patients with the hospital’s hot water system under the new energy supply.
Comfortable degree of indoor environment	*X* _7_	Refers specifically to temperature and humidity inside the hospital, reflecting the satisfaction of patients with the location of the hospital and the air-conditioning system under the new energy supply.
The level of ward management services	*X* _8_	A good level of ward management can speed up patient recovery and reduce their length of stay in hospital. This indicator reflects the satisfaction of the patients with the level of ward management services.
The appropriate quality of treatment	*X* _9_	Represents the appropriateness of the number of medical services received by patients. This indicator reflects the satisfaction of the patients with the appropriateness of medical behaviors.
The flow of people in the hospital	*X* _10_	Fewer people in the hospital can reduce carbon emissions and improve the patient’s medical experience. This indicator reflects the satisfaction of patients with the flow of people in the hospital.
Application of the digital platform	*X* _11_	The wise medical model can digitize some medical processes and directly reduce carbon emissions. This indicator reflects patients’ satisfaction with the construction of the hospital’s digital platform.
Catering system	*X* _12_	The operation of the catering system is also an important source of carbon emissions. The supply of new energy will have a certain impact on the production, insulation and transportation of catering. This indicator reflects the satisfaction of the patients with the hospital catering system.
Quality of telemedicine services	*X* _13_	Telemedicine service is an important component of wise medical. It can reduce patients’ travel and thus reduce carbon emissions. This indicator reflects patients’ satisfaction with the level of telemedicine services.
Cost of telemedicine services	*X* _14_	While telemedicine brings convenience, it naturally brings about other derived problems such as high price. This indicator reflects patients’ satisfaction with the reasonableness of telemedicine prices.

**Table 3 ijerph-19-04858-t003:** The adjustment results of parameters of the XGBoost model.

Parameter Name	Initial Value	Parameter Adjustment Range	Result
n_estimators	650	[700, 725, 750, 775, 800, 825]	825
min_child_weight	1.5	[1, 2, 3, 4, 5]	1
max_depth	5	[3, 4, 5, 6, 7, 8]	7
gamma	1	[0.2, 0.3, 0.4, 0.5, 0.6, 0.7]	0.4
subsample	0.7	[0.6, 0.7, 0.8, 0.9]	0.7
colsample_bytree	0.7	[0.6, 0.7, 0.8, 0.9]	0.7
reg_alpha	0	[0, 0.03, 0.05, 0.1, 1, 2]	1
reg_lambda	0	[0, 0.05, 0.1, 1, 2, 3]	1
learning_rate	0.2	[0.01, 0.03, 0.05, 0.07, 0.1, 0.15, 0.2]	0.2

**Table 4 ijerph-19-04858-t004:** Index values for performance evaluation of XGBoost models.

	Precision	Recall	f1-Score	Support
0	0.83	0.89	0.86	222
1	0.87	0.80	0.83	197
accuracy			0.85	419
macro avg	0.85	0.84	0.85	419
weighted avg	0.85	0.85	0.85	419

## Data Availability

The data that support the findings of this study are available from the corresponding author upon reasonable request.

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
