# Peer review of "Low-Carbon Action in Full Swing: A Study on Satisfaction with Wise Medical Development"

_ijerph, 2022, doi:10.3390/ijerph19084858_

Round 1

Reviewer 1 Report

Thank you for inviting me to review this interested paper “Low-carbon Action in Full Swing: a Study on the Satisfaction of Wise Medical Use” . After carefully review, I suggest some minor revision before publication.

1. A good article should include, (1) originality, new perspectives or insights; (2) international interest; and (3) relevance for governance, policy or practical perspectives relevant to the focus of this manuscript.

The authors need to clarify some questions: Why is the topic important (or why do you study on it)? What are research questions? What are your contributions? Why is to propose this particular method (This must come from Literature discussion)?

2. Your conclusions' section needs to underscore the scientific value added of your paper, and/or the applicability of your findings/results, as indicated previously. Basically, you should enhance your findings, limitations, underscore the scientific value added of your paper, and/or the applicability of your contributions/shortages and future study in this session.

Author Response

Thank you for inviting me to review this interested paper “Low-carbon Action in Full Swing: a Study on the Satisfaction of Wise Medical Use” . After carefully review, I suggest some minor revision before publication.

  1. A good article should include, (1) originality, new perspectives or insights; (2) international interest; and (3) relevance for governance, policy or practical perspectives relevant to the focus of this manuscript.

The authors need to clarify some questions: Why is the topic important (or why do you study on it)? What are research questions? What are your contributions? Why is to propose this particular method (This must come from Literature discussion)?

Response 1: Please provide your response for Point 1. (in red)

Thanks for your attentive and objective review, we have carefully read your comments and made careful revisions to the original manuscript after reflection. We further understood the characteristics of a good article and reviewed the existing related research, added a lot of literature discussion in the citation and discussion section, and put forward our views on the definition of wise medical and practical view of the later development of wise medical. Then, we would like to clarify the three issues that you have highlighted here. For the first three questions, we have added a lot of content to the introduction, and introduced the content of the background of COVID-19,further emphasizing the importance of our theme (P1-P3), and then led to our research questions and corresponding contributions. As for the particular method we proposed, we first discussed the feature of the method itself, and discussed the matching of the method we used ,including semantic co-word network (P4-P5), XGBoost (P5), etc., with our research topic. And secondly, in the Discussion section (P12-P14), we have added extensive literature to elaborate on the limitations of existing studies and the interpretation of our results.

  1. Your conclusions' section needs to underscore the scientific value added of your paper, and/or the applicability of your findings/results, as indicated previously. Basically, you should enhance your findings, limitations, underscore the scientific value added of your paper, and/or the applicability of your contributions/shortages and future study in this session.

Response 2: Please provide your response for Point 2. (in red)

Thanks for your comments. We have carefully reflected on the scientific value and applicability of our paper and made revisions. We have added a large number of existing research literature in the discussion section, discussed the similarities and differences between our research conclusions and other people's research results, expanded our thinking on the basis of other people's research results, and explained the value of our research results (on pages 12-14 of the revised manuscript).Regarding the limitations and future prospects of our research, we further added the limitations of data measurement and discussed the particularity of the study site in Zhejiang and the possibility of generalizing the results on the basis of the original analysis. In addition, under the current global background of the re-emergency of COVID-19, blockades have occurred in various places. At this time, the later research prospects can start from the heterogeneity of patients' demand for wise medical, and further explore how it can break the occlusion of information and the imbalance between supply and demand of medical resources in the epidemic environment.

Reviewer 2 Report

It is an interesting paper about the development of wise medical and it’s role to global carbon reduction. The study is original and it shows, that 837 valid questionnaires collected in Zhejiang Province, the XGBoost model is constructed, and the regional heterogeneity among the coastal area of eastern Zhejiang, the plain area of northern Zhejiang and the mountainous area of southwestern Zhejiang are discussed. The authors are using a selection of indicators affecting wise medical and the measurement of Indicators Affecting Wise Medical, and finally the Measurement of Level of Wise Medical.

It is an interesting paper, but the references are not relevant. My suggestion is to review and update the references. 

The paper presents its points in a rather descriptive manner, referring to relevant sources but never really discussing them.

The recommendations and suggestions are:

  1. To develop the section of “Conclusions” with the interpretation of results, the implications for further research and challenges of Industry.
  2. To update the references, to include more relevant reference for this topic.

The relationship between Low-carbon action, satisfaction of medical use and climate change has not been covered, and thus such recent sources should be cited:

Morrison, M., and Lăzăroiu, G. (2021). “Cognitive Internet of Medical Things, Big Healthcare Data Analytics, and Artificial intelligence-based Diagnostic Algorithms during the COVID-19 Pandemic,” American Journal of Medical Research 8(2): 23–36. doi: 10.22381/ajmr8220212.

Ionescu, L. (2021). “Transitioning to a Low-Carbon Economy: Green Financial Behavior, Climate Change Mitigation, and Environmental Energy Sustainability,” Geopolitics, History, and International Relations 13(1): 86–96. doi: 10.22381/GHIR13120218.

Pera, A. (2019). “Towards Effective Workforce Management: Hiring Algorithms, Big Data-driven Accountability Systems, and Organizational Performance,” Psychosociological Issues in Human Resource Management 7(2): 19–24. doi:10.22381/ PIHRM7220193

Riley, A., and Nica, E. (2021). “Internet of Things-based Smart Healthcare Systems and Wireless Biomedical Sensing Devices in Monitoring, Detection, and Prevention of COVID-19,” American Journal of Medical Research 8(2): 51–64. doi: 10.22381/ajmr8220214.

Walters, M., and Kalinova, E. (2021). “Virtualized Care Systems, Medical Artificial Intelligence, and Real-Time Clinical Monitoring in COVID-19 Diagnosis, Screening, Surveillance, and Prevention,” American Journal of Medical Research 8(2): 37–50. doi: 10.22381/ajmr8220213.

Author Response

The recommendations and suggestions are:

  1. To develop the section of “Conclusions” with the interpretation of results, the implications for further research and challenges of Industry.

Response 1: Please provide your response for Point 1. (in red)

Thanks for your meticulous review and comments, we have carefully considered your questions and made revisions one by one. In order to supplement the reasons for the results, challenges to future research and the industry, we add a lot of content in the two sections Discussion and Research Limitations to discuss the existing literature, and propose the impact of further research and future directions (P12- P14).Then, the interpretation part of the results is scattered in the middle analysis part (satisfaction impact analysis: P10-P11; regional heterogeneity discussion: P11-P12). After fully considered the epidemic background and geographical factors , we analyzed and explained the results, and compared it with the results of related research in the final discussion part, to further highlight the necessity of and the possible academic value of our research.

  1. To update the references, to include more relevant reference for this topic.

The relationship between Low-carbon action, satisfaction of medical use and climate change has not been covered, and thus such recent sources should be cited:

Morrison, M., and Lăzăroiu, G. (2021). “Cognitive Internet of Medical Things, Big Healthcare Data Analytics, and Artificial intelligence-based Diagnostic Algorithms during the COVID-19 Pandemic,” American Journal of Medical Research 8(2): 23–36. doi: 10.22381/ajmr8220212.

Ionescu, L. (2021). “Transitioning to a Low-Carbon Economy: Green Financial Behavior, Climate Change Mitigation, and Environmental Energy Sustainability,” Geopolitics, History, and International Relations 13(1): 86–96. doi: 10.22381/GHIR13120218.

Pera, A. (2019). “Towards Effective Workforce Management: Hiring Algorithms, Big Data-driven Accountability Systems, and Organizational Performance,” Psychosociological Issues in Human Resource Management 7(2): 19–24. doi:10.22381/ PIHRM7220193

Riley, A., and Nica, E. (2021). “Internet of Things-based Smart Healthcare Systems and Wireless Biomedical Sensing Devices in Monitoring, Detection, and Prevention of COVID-19,” American Journal of Medical Research 8(2): 51–64. doi: 10.22381/ajmr8220214.

Walters, M., and Kalinova, E. (2021). “Virtualized Care Systems, Medical Artificial Intelligence, and Real-Time Clinical Monitoring in COVID-19 Diagnosis, Screening, Surveillance, and Prevention,” American Journal of Medical Research 8(2): 37–50. doi: 10.22381/ajmr8220213.

Response 2: Please provide your response for Point 2. (in red)

Recognizing the relatively small number of our references, we re-searched and integrated relevant research materials, introduced them into our revised manuscript and discussed them. These new papers are mainly distributed in the Introduction and Discussion sections. Meanwhile,we have taken  full account of the recent sources you have recommended and used some of them. In the last,nearly 40 new papers have been added in total.

Reviewer 3 Report

The study is well performed and reported, with only a few exceptions:

First, the discussion of limitations and prospects is very short. You should elaborate more on 1) potential limitations of data, including selection or other bias issues 2) limitations in scope of the results, i.e., to which extent can the results be generalized to other cases and countries, and 3) where to move in future studies.

Second, I lack from the beginning some operational definition of the term "wise medical". It may be clear to the authors what it means, and it became clear to me through reading the paper, buit the general reader would be helped by a definition to steer reading from.

Author Response

First, the discussion of limitations and prospects is very short. You should elaborate more on 1) potential limitations of data, including selection or other bias issues 2) limitations in scope of the results, i.e., to which extent can the results be generalized to other cases and countries, and 3) where to move in future studies.

Response 1: Please provide your response for Point 1. (in red)

Thanks for your meticulous review, we have carefully considered your comments and made revisions one by one. First of all, in terms of the limitations and prospects of this paper, we added content about data limitations and the possibility of generalization of the results to the original manuscript, and discussed the potential in-depth research directions. We thought about the limitations of our data from the perspective of hospitals, and recognized the particularity of Zhejiang Province. We also included the geographic blockade brought by the COVID-19 pandemic into our later thinking and expanded our later research directions (P14).

Second, I lack from the beginning some operational definition of the term "wise medical". It may be clear to the authors what it means, and it became clear to me through reading the paper, buit the general reader would be helped by a definition to steer reading from.

Response 2: Please provide your response for Point 2. (in red)

Thank you very much for your additional comments on our paper, we recognize the importance of developing a definition of “wise medical” in the beginning. We reintroduced new research literature in the introduction, then discussed and supplemented the definition of wise medical to make up for this missing content. The new definition is on the second page of the manuscript.

Round 2

Reviewer 2 Report

It is an interesting paper, but the references are not relevant for the research. My suggestion is to update the references and to include more international literature, as I suggested in my first review. I suggested some international references, part of them are included in your paper.

Author Response

Response 1 :

We have revised the references of the manuscript once again, and all the international references you suggested are considered by us. Thank you for your comments.
